# Neurosymbolic models based on hybrids of convolutional neural networks and decision trees

**Rasul Kairgeldin**     **Miguel Á. Carreira-Perpiñán**
**Dept. of Computer Science & Engineering, University of California, Merced**
http://eecs.ucmerced.edu

**Editors:** Leilani H. Gilpin, Eleonora Giunchiglia, Pascal Hitzler, and Emile van Krieken

## Abstract

Building on previous work, we propose a specific form of neurosymbolic model consisting of the composition of convolutional neural network layers with a sparse oblique classification tree (having hyperplane splits using few features). This can be seen as a neural feature extraction that finds a more suitable representation of the input space followed by a form of rule-based reasoning to arrive at a decision that can be explained. We show how to control the sparsity across the different decision nodes of the tree and its effect on the explanations produced. We demonstrate this on image classification tasks and show, among other things, that relatively small subsets of neurons are entirely responsible for the classification into specific classes, and that the neurons' receptive fields focus on areas of the image that provide best discrimination.

## 1. Introduction

Neural AI systems and symbolic AI systems have both developed extensively since the mid-twentieth century and achieved impressive successes in different domains. Symbolic AI systems, which include many different models (rule-based, logic-based, etc.) are typically characterized by their transparent nature, in that a human can follow their chain of reasoning, at least to some extent. This brings control and trust in the results and often some form of correctness guarantees (which is critical in, say, program verification or theorem proving). Neural AI systems, whose most powerful form at present are deep neural nets (NNs), have excelled at perceptual tasks, such as recognizing patterns in images or generating realistic text. This is due to their ability to learn from large labeled datasets by optimizing an objective function that measures the prediction error over the NN parameters. This, in turn, is made possible by the fact that NNs define differentiable functions, whose gradient can be used in effective optimization algorithms such as SGD. This also makes it possible to combine multiple NNs and train them jointly (end-to-end), thanks to the chain rule of derivatives. On the other hand, neural AI systems typically require large computational resources in terms of hardware, energy, and training and test time and memory. Also, their reasoning is impenetrable to humans: even though the structure and parameters can be inspected, their size and complexity is such that they behave like a black box. This makes it hard to understand why they make mistakes and how to correct them, among other things.

Thus, there has long existed an interest in neurosymbolic systems (Hitzler and Sarker, 2021; Kautz, 2022; d'Avila Garcez and Lamb, 2023), which seek to combine the best of both worlds. In particular, there have been previous attempts to combine NNs and trees (see

section 2). Here, building on the recent work of Hada et al. (2024) (HCZ24 for short), we will focus on a hybrid model consisting of specific forms of neural and symbolic AI systems: convolutional NNs (although other NNs could also be used) and decision trees, respectively. The trees can be turned into a set of decision rules if so desired. The hybrid can be seen as having the CNN learn some complex features that represent the input (say, an image) in a way that is more suitable for classification; while the classification is effected by the decision tree, whose structure and logic are interpretable by a human (to some extent, depending on the size of the tree). For example, one can follow the tree reasoning by tracing the path followed by the input instance from the root to a leaf, solve counterfactual explanations exactly (Carreira-Perpiñán and Hada, 2021), etc.

Following HCZ24, their procedure starts from a trained CNN and replaces a part or module $\mathbf{M}$ of it with a decision tree that aims at being *approximately functionally equivalent (a.f.e.)*[1]. This is achieved by training the tree in a teacher-student way on a dataset with the inputs that $\mathbf{M}$ receives labeled with the corresponding outputs that $\mathbf{M}$ produces. For example, in a LeNet or VGG CNN, $\mathbf{M}$ could be all the fully-connected layers that follow the last convolutional layer (fig. 1). If the predictions of the tree are identical or very close to those of $\mathbf{M}$ on the training and test set, we deem it to be a.f.e. to $\mathbf{M}$. Then, by replacing $\mathbf{M}$ with the tree, we define a hybrid, neurosymbolic model that is a.f.e. to the original CNN. However, the symbolic module (the tree or set of rules) brings some amount of explainability into the hybrid model, which (since the tree represents $\mathbf{M}$ well) transfers to the original CNN. For example, among other things that HCZ24 explored is the fact that relatively small subsets of neurons are associated with specific classes. This makes it possible to manipulate the CNN with surgical precision to force it to make certain classification decisions. The hybrid model can also replace the original CNN for actual prediction but with much faster inference.

Key to the success of this approach is the ability of the tree to match the predictive performance of the module $\mathbf{M}$, while remaining sufficiently interpretable. This may not always be possible, depending on the complexity of $\mathbf{M}$, but recent advances in decision tree learning have greatly enlarged the space of tree-type models we can train and improved the accuracy they can achieve and their scalability to large, high-dimensional datasets. In particular, we will use *sparse oblique trees* for classification (Carreira-Perpiñán and Tavallali, 2018), which predict a constant label at each leaf, but use hyperplane splits with few features at each decision node. Such trees can be effectively trained with the Tree Alternating Optimization (TAO) algorithm (Carreira-Perpiñán and Tavallali, 2018). Indeed, as shown in HCZ24's and our experiments, the resulting tree is very small (often having just one leaf per class), yet it produces an a.f.e. hybrid to the original CNN. This would not be possible with the traditional CART-style decision tree induction algorithms, which use a much more limited tree type (axis-aligned, using a single feature per split) and a much more suboptimal training algorithm (greedy recursive partitioning).

In our paper, a first contribution is to make this specific type of model (which would be a "type-3 or Neuro|Symbolic" system in the classification of Kautz (2022)) known to

---

1. By this we do not mean the tree represents exactly the same mathematical function as $\mathbf{M}$. This much stricter requirement would be overkill in practice because real data occupy a tiny, low-dimensional region of the input space. An estimate of this region is given (indirectly) by the observed data we have for training and test.

the neurosymbolic learning community. A second contribution is that we modify the TAO algorithm so that it learns sparse oblique trees with a controllable sparsity distribution over the nodes, thus increasing their interpretability while remaining highly accurate. A third contribution is that we take the interpretability of the hybrid model beyond what HCZ24 did, by showing the receptive fields of neurons that are responsible for the discrimination between specific classes. We display this as a density map that objectively shows where in the image those neurons are looking. For example, in the Fashion MNIST dataset we show how certain neurons act on specific image areas to detect the presence of specific object parts that are critical to tell one class from another (e.g. to tell a shoe from a bag, a critical part is the existence of a gap in the shoe front which is occupied by a corner in the bag, and a certain neuron detects precisely that). Also, using this knowledge, we are able to edit an image that the CNN misclassifies so it classifies it correctly.

## 2. Related work

We focus on work involving tree- or rule-based models and neural nets (NNs). Early on, the black-box nature of NNs was recognized and there were attempts to replace the entire, trained NN with a symbolic representation, specifically a decision tree or a set of rules, which provided an explainable system. Several approaches of this type were actively researched in the 1990s and 2000s (reviewed in (Andrews et al., 1995; Duch et al., 2004; Jacobsson, 2005; McCormick et al., 2013; Guidotti et al., 2018)). One approach (Towell and Shavlik, 1993; Fu, 1994; Setiono and Liu, 1996; Baesens et al., 2003; Duch et al., 2004) relied only on access to the architecture and weight values of a multilayer perceptron (MLP), although the neuron activations were assumed to be binary. Rules were extracted using some form of heuristic search. The other is a teacher-student approach, then called "pedagogical", which needs a training set (actual or synthetic) on which a decision tree or a set of rules is trained to mimic the input-output behavior of the MLP (Craven and Shavlik, 1994, 1996; Domingos, 1998). Although the experiments in these papers were somewhat successful, they were limited to tiny two-layer MLPs and never scaled up to larger NNs (having more units and layers). This was due to the use of explainable models of limited power (such as axis-aligned trees) and/or to the heuristic nature of the procedure (a heuristic search or a suboptimal training with greedy recursive partitioning algorithms such as CART (Breiman et al., 1984) or C4.5 (Quinlan, 1993)). In contrast, the work of Hada et al. (2024) did scale up to larger NNs (LeNet, VGG), without any assumption on the type of activations or NN architecture other than the ability of the tree to match the accuracy of the NN module replaced. This was achieved by using a more powerful type of trees (oblique) and a better optimization (TAO), and also by aiming to replace a part of a NN rather than all of it.

Another related line of work is based on soft decision trees (SDTs) (Jordan and Jacobs, 1994), which use a sigmoid instead of step function at each decision node. Thus, an input instance traverses all root-leaf paths rather than a single one as in a hard tree, and the SDT output is a weighted average of all the leaves' labels. This defines a differentiable mapping which can be optimized via gradient-based methods, possibly end-to-end together with other modules (Kontschieder et al., 2015; Good et al., 2023; Hazimeh et al., 2020; Ibrahim et al., 2024; Borisov et al., 2024). Indeed, a SDT is much closer to (a specific type of) NNs than to trees. However, the fact that the input instance follows all paths, thus

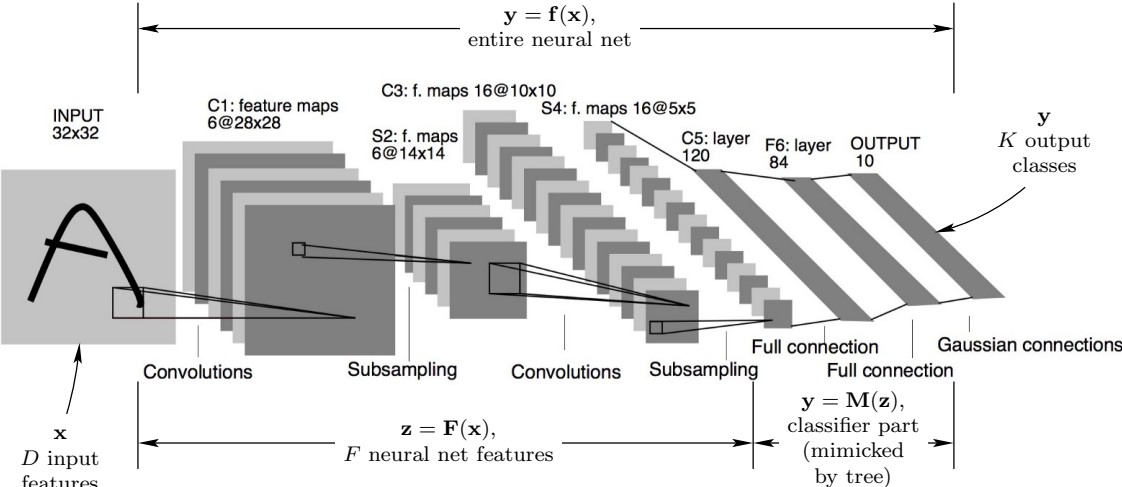

Figure 1: (Adapted from Hada et al. (2021, 2024).) A neurosymbolic model as a CNN-tree hybrid. The "neural feature" vector $\mathbf{z}$ consists of the activations (outputs) of the $F$ neurons in the last convolutional layer. The fully-connected MLP layers $\mathbf{M}$ are replaced with a sparse oblique classification tree.

touching all parameters, makes the SDT a black box. For the same reason, training and inference are also much slower than with a hard tree. One can turn a SDT into a hard tree by replacing each sigmoid with a step function, but this degrades the accuracy considerably and is worse than training a hard tree directly (Gazizov et al., 2025).

Finally, appending a tree to a neural net feature extraction has also been done for other purposes, such as ensembling (Zharmagambetov and Carreira-Perpiñán, 2021) or compression (Idelbayev et al., 2025).

## 3. The neural net / tree hybrid model: definition and training

We use the same basic model proposed in HCZ24 and illustrated in fig. 1. The starting point is a convolutional neural net (CNN), previously trained using a dataset (training and test) of labeled images (or other type of data) for a classification task. We regard the CNN as having the form $\mathbf{y} = \mathbf{M}(\mathbf{F}(\mathbf{x}))$, where $\mathbf{x}$ is the pixel image, $\mathbf{y}$ the predicted label (or label distribution), $\mathbf{F}$ the convolutional layers and $\mathbf{M}$ the fully-connected layers (i.e., an MLP). The partition of the original NN could be done in other ways, but this particular one makes sense in that the output of $\mathbf{F}$ (the output of the last convolutional layer, having $F$ neurons) can be seen as having learned a feature representation[2] or embedding, while $\mathbf{M}$ acts as a classifier on them.

Next, the fully-connected layers $\mathbf{M}$ are replaced by another classifier, namely a sparse oblique decision tree $\mathbf{y} = \mathbf{T}(\mathbf{z})$. This is done in a teacher-student way, by constructing a new dataset[3] having a pair $(\mathbf{z}_n, \mathbf{y}'_n)$ for every original data pair $(\mathbf{x}_n, \mathbf{y}_n)$, where $\mathbf{z}_n = \mathbf{F}(\mathbf{x}_n)$

---

2. One could also use non-adaptive features such as SIFT, constructed via a fixed formula, and interpretable by design.

3. If, instead of approximating the original NN, we seek the best possible hybrid model, we would use $\mathbf{y}'_n = \mathbf{y}_n$ instead (i.e., minimize the error on the ground-truth labels). In practice with NNs, especially

(the output of the last convolutional layer) and $\mathbf{y}'_n = \mathbf{M}(\mathbf{z}_n)$ (the output of the CNN). If we can train a tree $\mathbf{T}$ such that the error $\|\mathbf{y}'_n - \mathbf{T}(\mathbf{z}_n)\|$ is very small over the training and test data, then $\mathbf{T}$ and $\mathbf{T} \circ \mathbf{F}$ are a.f.e. to $\mathbf{M}$ and $\mathbf{M} \circ \mathbf{F}$, respectively. In HCZ24's and our experiments this is the case. The reason is that sparse oblique trees trained with TAO are quite powerful classifiers.

Finally, we obtain our hybrid CNN-tree model $\mathbf{y} = \mathbf{T}(\mathbf{F}(\mathbf{x}))$, which can be used in place of the original CNN. Besides providing faster inference, the tree makes it possible to understand the workings of the CNN features and we explore this in our experiments.

At present, we do not have a way to train the hybrid model end-to-end (i.e., $\mathbf{T}$ and $\mathbf{F}$ jointly). This because the tree defines a piecewise constant function, so its gradient is zero nearly everywhere in parameter space and the chain rule cannot be used to update $\mathbf{F}$. However, this is not a limitation if our goal is to understand the meaning and effect on classification of the original CNN features $\mathbf{F}$.

## 4. The Tree Alternating Optimization (TAO) algorithm: review

Due to space constraints, we keep this brief. More details can be found in the original references (Carreira-Perpiñán and Tavallali, 2018; Hada et al., 2024).

Tree Alternating Optimization provides a unified framework for effectively training complex decision tree-based models. We will discuss it in the setting of training one oblique decision tree. Consider a training set $\{(\mathbf{x}_n, \mathbf{y}_n)\}_{n=1}^N \subset \mathbb{R}^D \times \{1, \dots, K\}$ with $N$ samples, $D$-dimensional features, and $K$ classes. We define an oblique decision tree $\mathbf{T}(\mathbf{x}; \boldsymbol{\Theta})$ as a rooted binary tree with decision nodes $\mathcal{D}$ and leaf nodes $\mathcal{L}$. Each decision node $i \in \mathcal{D}$ employs a linear decision function $g_i(\mathbf{x}; \boldsymbol{\theta}_i)$ to route an instance $\mathbf{x}$ to the left (if $\mathbf{w}_i^T \mathbf{x} + w_{i0} \geq 0$) or right child (otherwise), where $\boldsymbol{\theta}_i = \{\mathbf{w}_i, w_{i0}\}$ are learnable parameters. Note how the decision function makes hard decisions, unlike in soft trees, where an instance $\mathbf{x}$ is propagated to both children with a positive probability. Each leaf $j \in \mathcal{L}$ contains a constant label classifier that outputs a single class $c_j \in \{1, \dots, K\}$. We collectively define the parameters of all nodes as $\boldsymbol{\Theta} = \{(\mathbf{w}_i, w_{i0})\}_{i \in \mathcal{D}} \cup \{c_j\}_{j \in \mathcal{L}}$. The predictive function of the whole tree $\mathbf{T}(\mathbf{x}; \boldsymbol{\Theta})$ then works by routing an instance $\mathbf{x}$ to exactly one leaf through a root-leaf path of (oblique) decision nodes and applying that leaf's predictor function.

Given a fixed-structure oblique decision tree $\mathbf{T}(\mathbf{x}; \boldsymbol{\Theta})$ (e.g. a complete tree of depth $\Delta$ or one from CART) with random initial parameters, TAO aims to minimize the following objective:

$$E(\boldsymbol{\Theta}) = \sum_{n=1}^N L(\mathbf{y}_n, \mathbf{T}(\mathbf{x}_n; \boldsymbol{\Theta})) + \lambda \sum_{i \in \mathcal{D}} \|\mathbf{w}_i\|_1 \tag{1}$$

where $L(\cdot, \cdot)$ is the loss (cross-entropy, 0/1, squared error, etc.) and the term with the hyperparameter $\lambda \geq 0$ is an $\ell_1$ penalty to promote sparsity of the weight vectors.

The TAO algorithm relies on two key theorems. The *separability condition* ensures that the objective function separates over non-descendant nodes (e.g. all nodes at a given depth), allowing for independent and parallel optimization over parameters of each node. The *reduced problem (RP) over a node* states that optimizing the objective for a node $i \in \mathcal{D} \cup \mathcal{L}$

---

if overparameterized, this makes little difference because the original NN usually has a very small error on the training set, so $\mathbf{y}'_n = \mathbf{y}_n$ for most training points.

simplifies to a well-defined problem involving only the training instances reaching that node (the *reduced set (RS)*, $\mathcal{R}_i \subset \{1, \ldots, N\}$). We define the following reduced problems. The exact form of the reduced problem differs for leaves and for decision nodes:

- For a decision node $i \in \mathcal{D}$, the top-level problem of eq. (1) reduces to a *weighted 0/1 loss binary classification problem*:

$$E_i(\mathbf{w}_i, w_{i0}) = \sum_{n \in \mathcal{R}_i} \overline{L}(\overline{y}_n, g_i(\mathbf{x}_n; \mathbf{w}_i, w_{i0})) + \lambda \|\mathbf{w}_i\|_1. \tag{2}$$

  Here, $\overline{y}_n \in \{\texttt{left}_i, \texttt{right}_i\}$ is a pseudolabel indicating the optimal child for $\mathbf{x}_n$, minimizing the subtree loss. The weighted 0/1 loss $\overline{L}(\cdot, \cdot)$ is defined by the loss difference between the chosen and alternative child. While optimizing an oblique node is generally NP-hard, it can be effectively approximated using a surrogate loss like cross-entropy (i.e., logistic regression). The top-level objective (1) is guaranteed to decrease by accepting updates only if they improve (2), though this is often unnecessary in practice.

- For leaf node $j \in \mathcal{L}$, the top-level problem of eq. (1) reduces to a form involving the original loss but only over the parameters of the leaf predictor function. It can be solved by finding the majority class (or mean value of the samples in the reduced set for regression)

While these theorems do not prescribe the order in which the nodes should be optimized, we follow a reverse breadth-first search order: all the nodes at a given depth are optimized in parallel, starting from the deepest ones until the root. Each optimization subproblem involves solving either an $\ell_1$-regularized logistic regression or finding a majority class. By ensuring that the (approximate) solution of the reduced problem of a decision node improves upon the previous node parameter values, TAO is guaranteed to decrease the objective function (1) monotonically.

Finally, node pruning occurs automatically because the $\ell_1$ penalty can drive a node's entire weight vector to zero. This makes the node redundant (it sends all instances to the same child) and it can be removed at the end.

## 5. Finer sparsity control with a modified TAO algorithm

The hyperparameter $\lambda$ controls the overall sparsity in the tree, and by making it large enough we also achieve pruning (and thus a form of tree structure learning) automatically. However, it also has the effect that shallow nodes (e.g. the root) are much less sparse than deeper nodes (e.g. the leaf parents). This is seen in the trained trees and its cause is explained below. We address this here with a second hyperparameter $\alpha$ that controls *how sparse individual nodes are* in relation to the number of instances they handle.

Consider the following objective function, equal to (1) but with a modified regularization term (whose motivation is explained later) with hyperparameter $\alpha \in \mathbb{R}$:

$$E(\boldsymbol{\Theta}) = \sum_{n=1}^{N} L(\mathbf{y}_n, \mathbf{T}(\mathbf{x}_n; \boldsymbol{\Theta})) + \lambda \sum_{i \in \mathcal{D}} h_\alpha(|\mathcal{R}_i|) \|\mathbf{w}_i\|_1, \qquad h_\alpha(t) = \begin{cases} 1, & t = 0 \\ t^\alpha, & t > 0 \end{cases} \tag{3}$$

where $\mathcal{R}_i$ is the RS of node $i$ and $|\mathcal{R}_i|$ its cardinality. This seems difficult to optimize: the $h_\alpha$ term is a non-differentiable function of the tree parameters (specifically, the weight

vectors of the decision nodes in the path ascending from $i$ to the root), because it depends on $|\mathcal{R}_i|$ (an integer), which depends on the said parameters. However, the TAO theorems still apply, and the only change is in the RP over a decision node $i$, which now takes the following form:

$$E_i(\mathbf{w}_i, w_{i0}) = \sum_{n \in \mathcal{R}_i} L(\mathbf{y}_n, \mathbf{T}(\mathbf{x}_n; \boldsymbol{\Theta})) + \lambda\, h_\alpha(|\mathcal{R}_i|)\, \|\mathbf{w}_i\|_1. \tag{4}$$

However, $\mathcal{R}_i$ is constant if the nodes ascending from $i$ are kept fixed (as they are in each TAO iteration). Thus, the RP is exactly as in TAO but with a reweighted hyperparameter "$\lambda\, h_\alpha(|\mathcal{R}_i|)$".

The reason for this new algorithm can be seen by dividing[4] the RP objective function by the constant $N_i = |\mathcal{R}_i|$ (the number of points in $i$'s RS) and rewriting it as "avg-loss + $\lambda'$ reg", where we define avg-loss $= \frac{1}{N_i} \sum_{n \in \mathcal{R}_i} L(\mathbf{y}_n, \mathbf{T}(\mathbf{x}_n; \boldsymbol{\Theta}))$ (the loss per instance in node $i$), reg $= \|\mathbf{w}_i\|_1$, and $\lambda' = \lambda N_i^{\alpha-1}$ (an *effective sparsity* hyperparameter). This makes it clear that $\alpha < 1$ (e.g. $\alpha = 0$ as in regular TAO) penalizes larger RSs *less* than smaller ones (thus the root weight vector is less sparse); $\alpha > 1$ penalizes larger RSs *more* than smaller ones; and $\alpha = 1$ penalizes all nodes equally, regardless of how many instances they receive. This gives us control on how the feature sparsity is distributed across the tree (clearly seen in fig. 2 vs fig. 7), which is useful for interpretability purposes. Further, it can actually find trees that are *both sparser overall and possibly even more accurate than those of the regular TAO algorithm.* Indeed, experimentally we observe that the trees with best generalization error occupy a relatively wide region in $(\lambda, \alpha)$ space, from which we can pick the sparsest tree.

## 6. Experiments

In this section, we experimentally demonstrate that our approach learns trees with higher node sparsity while maintaining, or even improving, accuracy. The resulting tree performs comparably to neural network. This, along with other experiments, suggests that insights gained from the tree also apply to the neural network. Importantly, our findings are guaranteed to be correct for hybrid model and verified to hold well empirically for the original NN; unlike those of other interpretability attempts based on saliency maps or Shapley values.

**Predictive error and tree size** We train LeNet-5 on the Fashion MNIST dataset using PyTorch 1.10. Full hyperparameters and TAO implementation details are provided in the Appendix. Our trees are trained on embeddings from the last convolutional layer ($F = 400$) to closely match NN performance. To maintain interpretability, we restrict tree depth to 5. The best-performing tree that uses only $1\,298$ non-zero parameters, achieving $E_{\text{train}} = 5.4\%$ and $E_{\text{test}} = 11.7\%$ is achieved with uniform sparsity distribution ($\lambda = 0.001, \alpha = 1$). In comparison, the NN's fully connected layers contain $59\,134$ parameters—nearly 50 times more—while improving error rates by only about $1\%$ on both train and test sets.

We also trained axis-aligned trees using CART and TAO. The best CART tree achieved a training error of $8.7\%$ and a test error of $23.4\%$ with a depth of 31 and $5\,567$ nodes. The best TAO univariate tree had a test error of $21.8\%$ with $4\,400$ nodes. These results highlight

---

4. This assumes $|\mathcal{R}_i| > 0$. If $\mathcal{R}_i = \varnothing$ then $h_\alpha(|\mathcal{R}_i|) = 1$ and the RP solution is to set $\mathbf{w}_i = \mathbf{0}$, as in TAO.

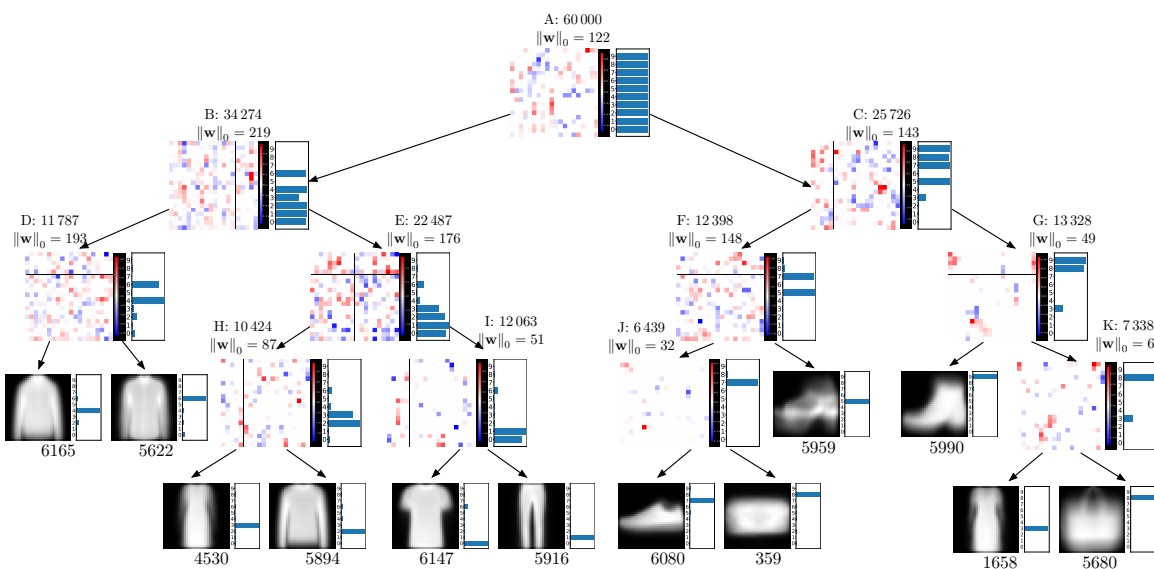

Figure 2: Tree trained on LeNet embeddings on Fashion MNIST dataset $\lambda = 0.001, \alpha = 1$. $E_{\text{train}} = 5.4\%$, $E_{\text{test}} = 11.7\%$, number of non-zero parameters is 1298.

the limitations of axis-aligned trees in capturing the complex feature interactions learned by neural networks while also losing interpretability due to their excessive size.

We further analyze the impact of sparsity parameters $\lambda$ and $\alpha$. Figure 10 illustrates the regularization path by fixing $\lambda$ while gradually increasing $\alpha$. As predicted by Eq. 4, increasing $\alpha$ enhances relative sparsity in decision nodes closer to the root. Beyond a certain threshold, the root becomes too sparse to sustain an oblique split, causing the tree to collapse. Lowering $\lambda$ shifts this threshold, allowing the tree to maintain its structure over a broader range of $\alpha$. This effect is evident in the figure: for $\lambda = 100$, $\alpha = 0.5$ leads to collapse, whereas for $\lambda = 1$, $\alpha = 0.5$ still preserves competitive performance. We find that there is a range of $\lambda$, $\alpha$ values that achieve best trees but with different sparsity distribution, from which we can pick the best one. For example, comparing 3 trees shows how sparsity distribution changes from uniform in Fig. 2 to closer to leaves in Fig. 6 and Fig. 7.

**Global structure** Figure 2 visualizes the structure of the best-performing tree, showing the learned sparse weights at decision nodes. Each node displays weights in a $4 \times 4$ grid, where each cell represents a $5 \times 5$ activation from the last convolutional layer of LeNet ($16 \times 5 \times 5$). Color coding indicates weight values: red for positive, blue for negative, and white for zero. At the top of each decision node and the bottom of each leaf, we display the number of non-zero weights and the size of the reduced set. Additionally, class histograms are shown on the right side of each node to illustrate the label distribution.

The tree has just 12 leaves, nearly one per class, significantly enhancing interpretability. Analyzing the leaf nodes reveals a clear hierarchical structure among the classes. The root node A utilizes only 122 activations to almost perfectly separate classes $\{9, 8, 7, 5\}$ from $\{6, 4, 2, 1, 0\}$, with minor misclassifications in class 3. Similarly, decision node C maintains comparable sparsity to the root while effectively distinguishing between $\{7, 5\}$ and $\{9, 8, 3\}$. Subtree at F specializes in distinguishing shoe types, and 359 samples (2.89% of the whole reduced set) of class 8 ("bag"). Interestingly, it uses 148 activations to distinguish class

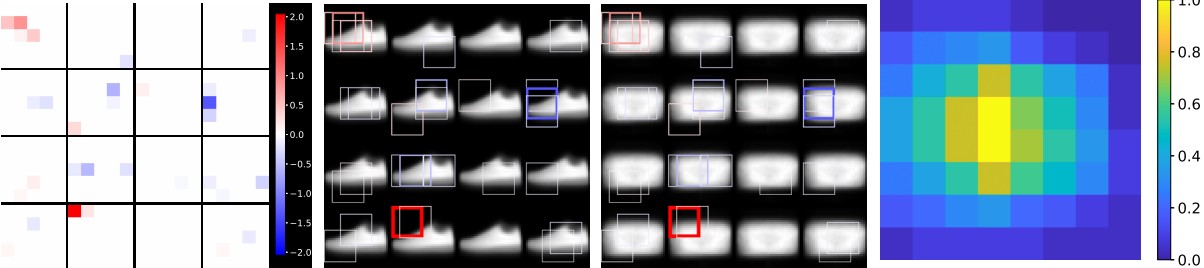

Figure 3: Where is a decision node looking at in the image? Plot 1 shows the sparse weights of decision node J for each CNN feature map in the last layer of LeNet ($16 \times 5 \times 5$). Plots 2–3 show the receptive field produced by neurons with non-zero decision node weights on the mean image from left and right leaf. Receptive fields follow the order of CNN outputs left to right and top to bottom. Color and thickness of receptive fields correspond to weights of decision node. Plot 4 is a heatmap of the "density" of the receptive fields. Tree hyperparameters: $\lambda = 0.001$, $\alpha = 1$.

5 ("sandal") from both "bags" and "sneakers", while differentiating between "sneakers" and "bags" uses relatively small subset of neurons (only 32 out of 400). Similarly, subtree G focuses on distinguishing classes $\{9, 8, 3\}$. Unlike F, subtree G) uses significantly fewer activations in decision nodes G (49) and K (66). These examples highlight how certain neurons specialize in recognizing specific patterns, allowing the sparse tree to retain only essential activations for classification. In contrast, decision node D requires 193 activations to differentiate class 4 ("coat") from class 6 ("shirt"). Subtree E is similar to subtree F as it uses more information to separate sets of classes ($\{3, 2\}$ and $\{0, 1\}$), but distinguishing within these sets requires far fewer information. Visual inspection suggests that the high similarity between these classes forces the tree to use more activations to maintain accuracy.

**Where is a specific neuron, and a specific decision node, looking at?** Although understanding the precise meaning of a neural feature is difficult, due to the complexity of the function of the pixels it defines, what is possible is to construct its *receptive field (RF)*. This is the region of the image that a neuron in a convolutional layer is looking at. It occurs by design: a neuron in convolutional layer $i$ only receives input from a small subset of neurons in layer $i - 1$. Also, since those neurons are organized spatially in a systematic way over the image grid, so are their RFs. We can construct the RFs for all $F$ neurons and then look at how they participate in a decision node. Fig. 3 shows the RFs for a selected node, with a thickness proportional to their weight in the node, overlaid on an input image. Interestingly, RFs associated with the highest positive weights are concentrated in the top left region of the image. This suggests that neurons identifying ink presence in this area play a crucial role in distinguishing between a bag and a shoe, as the shoe image typically lacks content in that specific RF location.

Also, for any decision node, we can construct a *RF density map* over the image as a linear combination of *all* the $F$ RFs (considered as 0/1 indicator functions over the image) using the magnitude of their weights in the node. This objectively indicates the region of the image that is being used in that node: zero density means it is not used at all, and the larger the density the more it is used (because many neurons look there and/or their

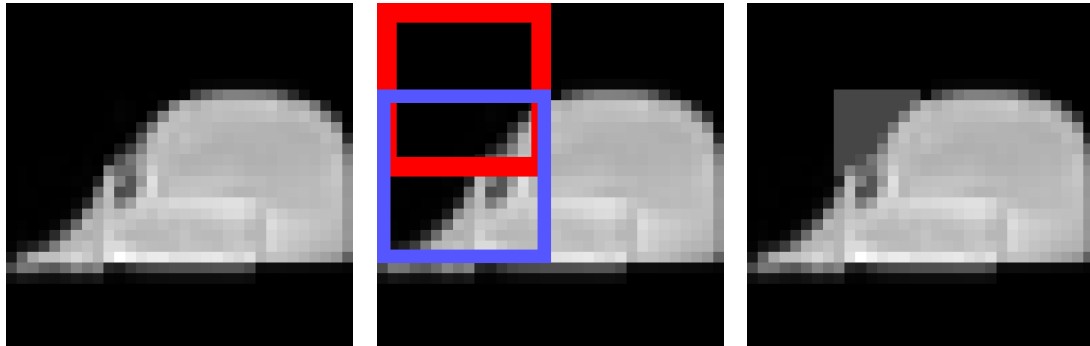

Figure 4: Sample of class 8 misclassified as 7 (Left). Receptive field of neurons from the last convolutional layer of LeNet with largest positive (red) and negative (blue) weights in oblique decision node J (Middle). Small changes in the intersection of two regions fixed the misclassification error (Right).

weights are large). In the example in fig. 3, this clearly shows that that node focuses on the region where typically we find either the hollow in the front of a shoe or the top-left corner of a bag, which (at that point in the tree) is sufficient to discriminate those two classes.

We emphasize that 1) this would not be possible without the tree and the weight vectors, and 2) it is much more objective than saliency maps and other NN visualization techniques that have been shown to be often misleading (Fong and Vedaldi, 2017; Adebayo et al., 2018; Ghorbani et al., 2019).

**Correcting classification mistakes of the neural net**  The above information can be used to understand why an image is (mis)classified as a certain class, and even to alter this by editing the image. Fig. 4 shows this with the image of a bag that the NN (and the tree) misclassifies as a shoe. From the RF density map discussed above it seems like the bag, which has an odd shape, is missing the typical left corner that bags have. When we edit the image to add that, both the NN and the tree classify it correctly.

## 7. Conclusion

We have revisited, and introduced to the neurosymbolic learning community, a hybrid model consisting of a convolutional feature extraction module composed with a sparse oblique decision tree. By introducing a new type of regularization and suitably modifying the Tree Alternating Optimization (TAO) algorithm, we have further developed this model to control how the feature sparsity is distributed over the tree structure. We train it in two stages: first, we train a regular CNN using SGD; then, we train the tree to replace its fully-connected layers using a teacher-student approach and our modified TAO algorithm. This produces an accurate hybrid model that benefits from the ability of convolutional layers to learn a better representation of an image, and from the ability of trees to explain the reasoning used to classify the image based on those neural features. This makes it possible to understand to some extent which neurons affect which classes, where in the image those neurons are looking at, why a specific image is (mis)classified as a certain class, and how to edit an image to alter its classification in desired ways.

## Acknowledgments

Work partially supported by NSF award IIS–2007147.

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

## Appendix A. Appendix

### A.1. Setup

We implemented TAO in Python 3.11. To solve reduced problem in the decision nodes we used `scikit-learn` logistic regression (Pedregosa et al., 2011) with LIBLINEAR (Fan et al., 2008). The regularization path was constructed by initializing tree with high different value of $\lambda$ and increasing $\alpha$ from negative values (typically $-1$) to 1 with a small step. We implemented LeNet in Pytorch 1.10. We trained it with Adam optimizer with learning rate of 0.001, and weight decay $1e-4$ on NVIDIA TITAN X. All tree experiments were conducted on the machine Intel Xeon CPU E5-2699 v3 @ 2.30GHz, 256 GB RAM.

### A.2. Masking

Fig. 9 was produced similar to HCZ24 by extracting a mask of non-zero entries from the sparse weight vector, then multiplying it element-wise with the activations from the last CNN layers. The result is passed through fully connected layer to produce classification. More details are described by Hada et al. (2024). The results show that the tree is able to capture specialized neurons, by keeping which, NN is only able to classify some specific classes.

---

**input** training set $\{(\mathbf{x}_n, y_n)\}_{n=1}^N \subset \mathbb{R}^D \times \{1, \ldots, K\}$
      initial tree $T$
      hyperparameters $\lambda \geq 0$, $\alpha \in \mathbb{R}$
**repeat**
  **for** $i \in$ nodes of $T$, visited in reverse BFS
    **if** $i$ is a leaf **then**
      $\theta_i \leftarrow$ majority-class label in the reduced set $\mathcal{R}_i$
    **else**
      generate pseudolabels $\overline{y}_n$ for each instance $n \in \mathcal{R}_i$
      $(\mathbf{w}_i, w_{i0}) \leftarrow$ minimizer of the reduced problem (eq. (4))
**until** stop
postprocess $T$: remove dead branches & pure subtrees
**return** $T$

Figure 5: (Adapted from Hada et al. (2021, 2024).) Pseudocode for the tree alternating optimization (TAO) algorithm, modified to handle our new sparsity regularization term with hyperparameter $\alpha \in \mathbb{R}$. Visiting each node in reverse breadth-first search (BFS) order means scanning depths from depth$(T)$ down to 1, and at each depth processing (in parallel, if so desired) all nodes at that depth. "stop" occurs when either the objective function decreases less than a set tolerance or the number of iterations reaches a set limit.

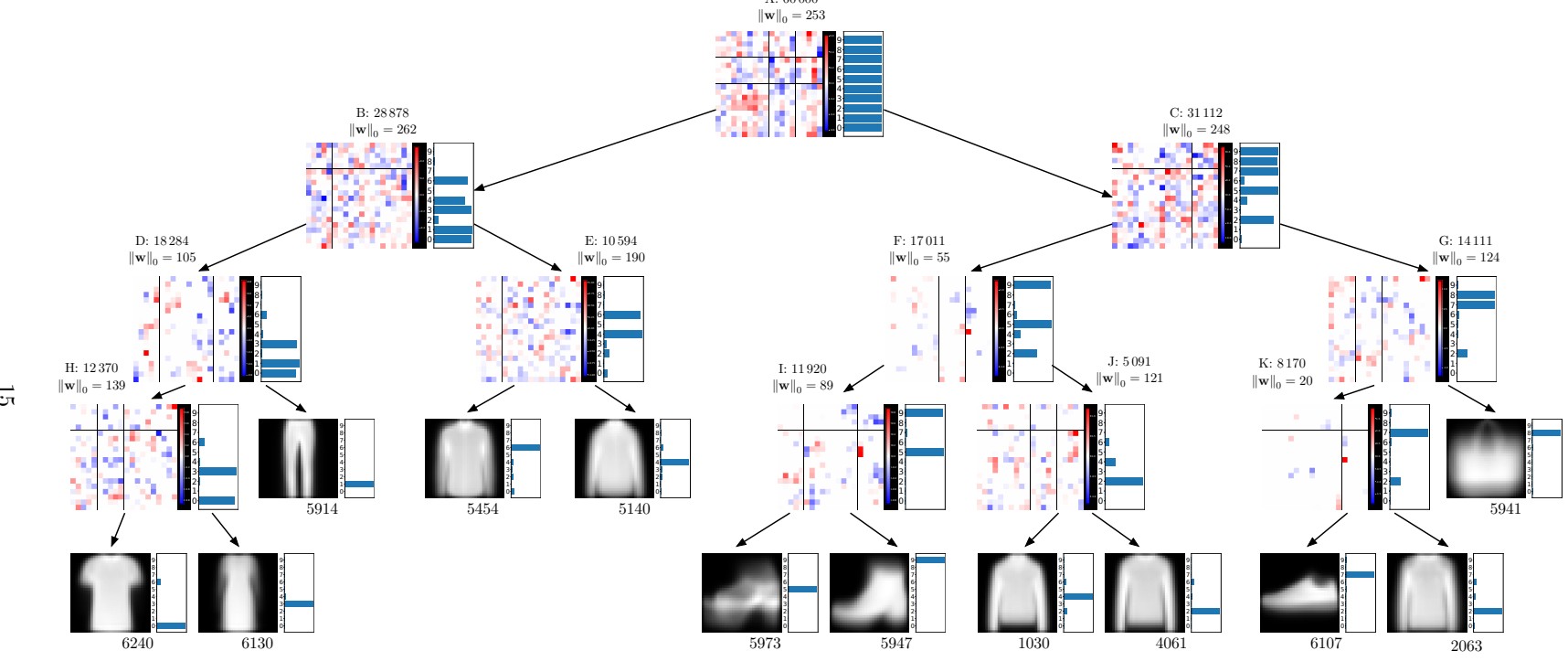

Figure 6: Tree trained on LeNet embeddings on Fashion MNIST dataset $\lambda = 1, \alpha = 0.25$. $E_{\text{train}} = 4.4\%$, $E_{\text{test}} = 12.4\%$, number of non-zero parameters is 1606.

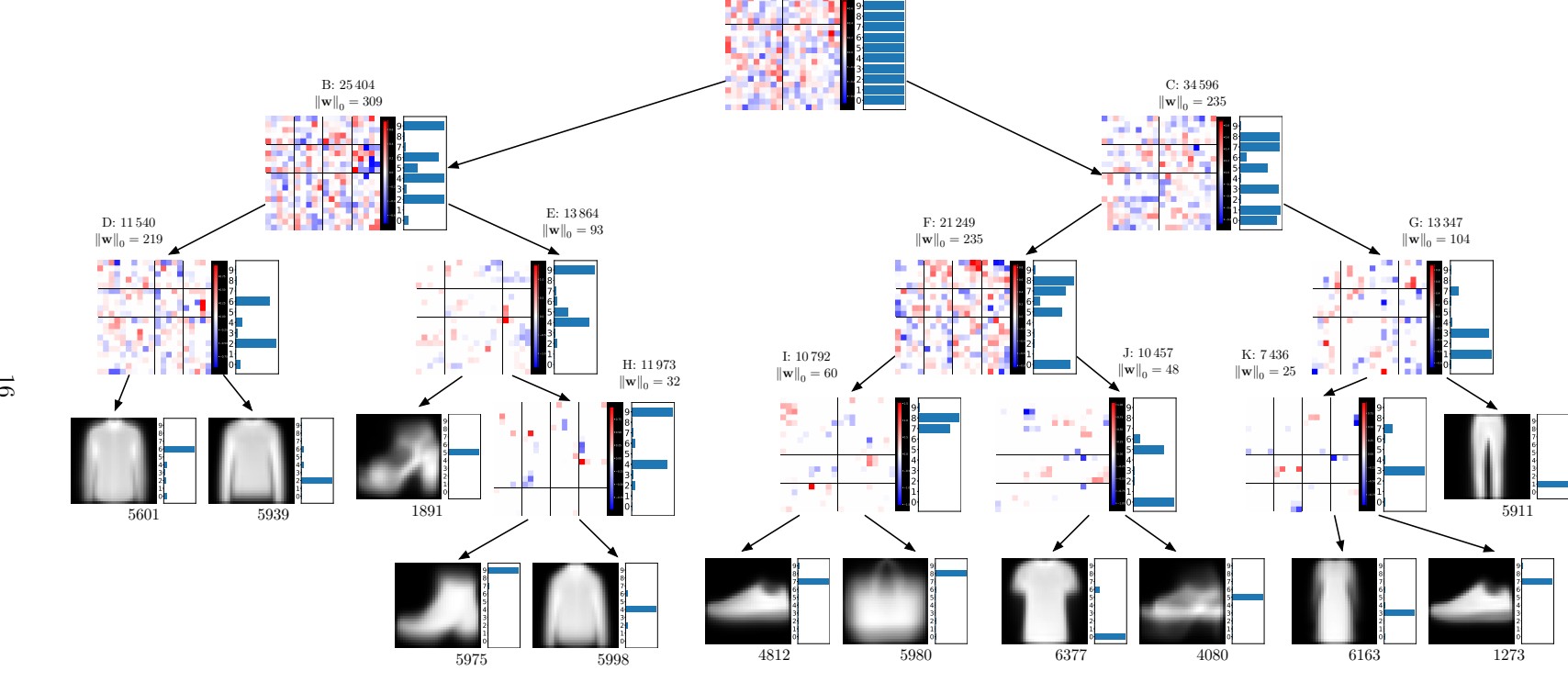

Figure 7: Tree trained on LeNet embeddings on Fashion MNIST dataset, $\lambda = 100, \alpha = -0.25$. $E_{\text{train}} = 5.2\%$, $E_{\text{test}} = 13.0\%$, number of non-zero parameters is 1701.

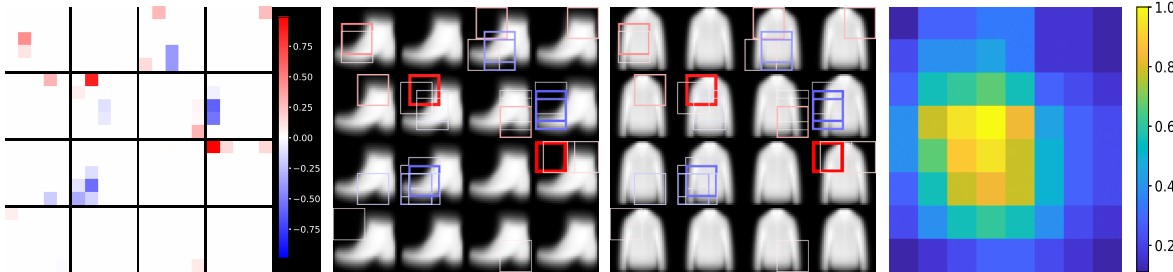

Figure 8: Where is a decision node looking at in the image? Plot 1 shows the sparse weights of decision node H for each CNN feature map in the last layer of LeNet ($16 \times 5 \times 5$). Plots 2–3 show the receptive field produced by neurons with non-zero decision node weights on the mean image from left and right leaf. Receptive fields follow the order of CNN outputs left to right and top to bottom. Color and thickness of receptive fields correspond to weights of decision node. Plot 4 is a heatmap of the "density" of the receptive fields. Tree hyperparameters: $\lambda = 100$, $\alpha = -0.25$.

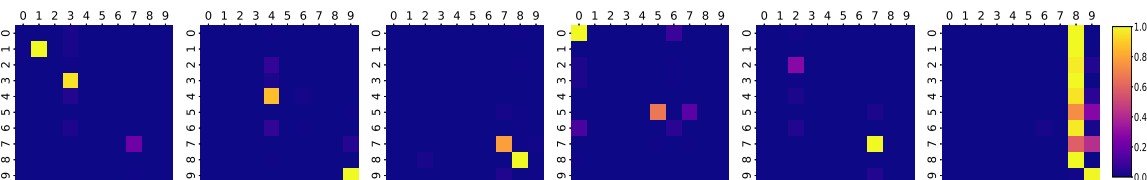

Figure 9: Confusion matrix of neural network predictions modified by tree mask. The masks are produced by analyzing sparse weights of decision nodes. From left to right first 4 are decision nodes G, H, I, J of tree with $\lambda = 100$, $\alpha = -0.25$. Next, decision node K of tree with $\lambda = 1$, $\alpha = 0.25$. Lastly, positive weights from subtree G in tree with $\lambda = 0.001$, $\alpha = 1$. The values are normalized.

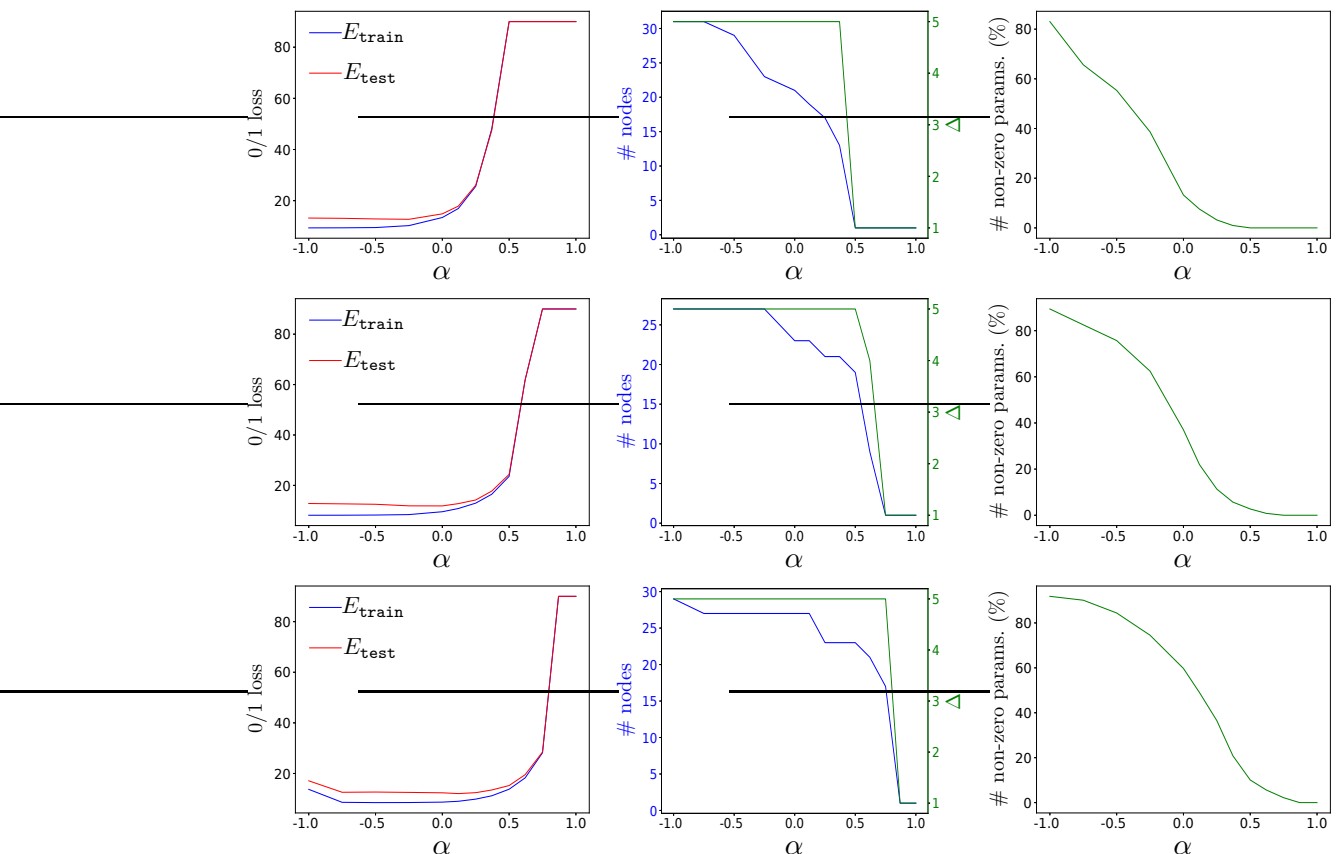

Figure 10: Regularization path over $\alpha$ for different $\lambda$ (top to bottom $\{100, 10, 1\}$).

