# OpenReview forum: "Neurosymbolic models based on hybrids of convolutional neural networks and decision trees"
_nesyconf.org/NeSy/2025/Conference — NeSy 2025 Poster_

### Official Review · Reviewer_PC96 · 2025-03-31
**Neurosymbolic Approach Combining CNNs and Sparse Decision Trees**

**Rating:** 7
**Confidence:** 3

**Review:**

Summary:
The paper proposes a neurosymbolic model that integrates convolutional neural networks (CNNs) with sparse oblique decision trees to enhance explainability while maintaining high classification accuracy. The approach is validated on an image classification task, demonstrating effective feature extraction and interpretability.

Review:
The paper introduces a hybrid model combining the strengths of CNNs in feature extraction and the interpretability offered by decision trees. Specifically, it leverages sparse oblique trees trained through the Tree Alternating Optimization (TAO) algorithm. The authors further propose a modification to TAO that allows finer control over sparsity, significantly enhancing interpretability without sacrificing accuracy.
The manuscript is well-written and structured and the paper’s contribution in refining sparsity control is original and practically beneficial.
The explicit visualization of neuron receptive fields and their contribution to interpretability is innovative and valuable.

Pros:
1. Provides a clear, interpretable method to partially analyze CNN decisions.
2. The modified TAO algorithm significantly enhances sparsity control.
3. Very interesting results on looking at particular neurons and demonstration of interpretability by correcting CNN misclassifications.

Cons:
1. A major formatting error is that references are in the wrong format. The paper is already 10 pages long and using the correct format will pass over this maximum, which leads to necessary rewriting of the paper. Yet the content is solid and this should be manageable.
2. Experiments primarily focus on Fashion MNIST, which is relatively simple; evaluation on more complex datasets would strengthen claims.

**Anonymity:**

Remain anonymous

---

### Official Review · Reviewer_Q6aZ · 2025-04-04
**Interesting paper, accept**

**Rating:** 9
**Confidence:** 3

**Review:**

This paper presents to the neurosymbolic community a hybrid model consisting of convolutional neural network layers and a sparse oblique decision tree. Specifically, this model is further developed by introducing a new type of regularization and modifying the Tree Alternating Optimization (TAO) algorithm to control how the feature sparsity is distributed over the tree structure. The hybrid model benefits from both the ability of convolutional layers to learn a better representation of an image and the ability of trees to explain the reasoning used to classify the image based on those neural features. A further contribution of the work to the interpretability of the model is to show the receptive fields of neurons responsible for discriminating between specific classes.

The paper provides interesting contributions and is very well written and presented. Therefore, I support its acceptance.

Below are some corrections/improvements:
- On page 3, in the fifth line from the bottom, "than" should be "that".
- In Section 4, the sets N_dec and N_leaf should be D and L, respectively.
- In section 4, explain that l_1 is the regularization term and indicate it in formula (1).
- In section 4, explain what "sparsity" means.
- On page 6, in the fourth line of the third paragraph, there is an "an" to be deleted.
- In Section 5, explain what "dense node" means.
- In Section 5, define the acronyms RS and RP before using them.
- In the subsection "Predictive error and tree size", 1298 non-zero parameters should be 1286.

**Anonymity:**

Remain anonymous

---

### Official Review · Reviewer_niq1 · 2025-04-07
**Combining CNNs with Tree Alternating Optimization**

**Rating:** 6
**Confidence:** 3

**Review:**

In the paper, a combination of CNN image classification with a decision tree approach is presented and evaluated with Fashion MNIST. The proposed method is a modified algorithm based on the already published Tree Alternating Optimization algorithm (TAO). The novel addition is a hyperparameter (alpha) to control how sparse indivdual nodes are in relation to the number of covered instances.
Combining blackbox  CNN image classification with a symbolic approach is a highly relevant topic for research on neuro-symbolic machine learning. The presented method TAO is interesting and technically sound. However, the contribution of this paper is not highly original -- it is a small (but interesting) add-on to the already published approach. The empirical evaluation has been done only with one data set (Fashion MNIST). It would be interesting to see how the approach works with richer images (e.g. a subset of ImageNet). I would suspect that the resulting trees would be much more complex (and less helpful). The presented evaluation results are rather explorative in character. A more extensive comparison of the effects of different hyperparameter settings with respect of tree size would have been interesting.

**Anonymity:**

Remain anonymous